# Medal S: Spatio-Textual Prompt Model for Medical Segmentation

Pengcheng Shi[1], Jiawei Chen[1,4], Jiaqi Liu[1],
Xinglin Zhang[1]†, Tao Chen[1,3], and Lei Li[1,2(✉)]

[1] Medical Image Insights, Shanghai, China
[2] University of Washington, Seattle, WA, USA
[3] University of Waterloo, Waterloo, ON, Canada
[4] Xi'an Jiaotong University, Xi'an, China
xinglinzh@gmail.com
lenny.lilei.cs@gmail.com

**Abstract.** We introduce Medal S, a medical segmentation foundation model that supports native-resolution spatial and textual prompts within an end-to-end trainable framework. Unlike text-only methods lacking spatial awareness, Medal S achieves channel-wise alignment between volumetric prompts and text embeddings, mitigating inaccuracies from resolution mismatches. By preserving full 3D context, it efficiently processes multiple native-resolution masks in parallel, enhancing multi-class segmentation performance. A lightweight 3D convolutional module enables precise voxel-space refinement guided by both prompt types, supporting up to 243 classes across CT, MRI, PET, ultrasound, and microscopy modalities in the BiomedSegFM dataset. Medal S offers two prompting modes: a text-only mode, where model predictions serve as spatial prompts for self-refinement without human input, and a hybrid mode, incorporating manual annotations for enhanced flexibility. For 24-class segmentation, parallel spatial prompting reduces inference time by more than 90% compared to sequential prompting. We propose dynamic resampling to address target-patch ratio imbalance, extending SAT and nnU-Net for data augmentation. Furthermore, we develop optimized text preprocessing, a two-stage inference strategy, and post-processing techniques to improve memory efficiency, precision, and inference speed. On the five-modality average on the validation set, Medal S outperforms SAT with a DSC of 75.44 (vs. 69.83), NSD of 77.34 (vs. 71.06), F1 of 38.24 (vs. 24.88), and DSC TP of 65.46 (vs. 46.97). Medal S achieves excellent performance by harmonizing spatial precision with semantic textual guidance, demonstrating superior efficiency and accuracy in multi-class medical segmentation tasks compared to sequential prompt-based approaches. Medal S will be publicly available at https://github.com/yinghemedical/Medal-S.

**Keywords:** Medical Segmentation · Foundation Model · Spatial and Textual Prompts.

---

[1] †Project Lead

[2] (✉)Corresponding author

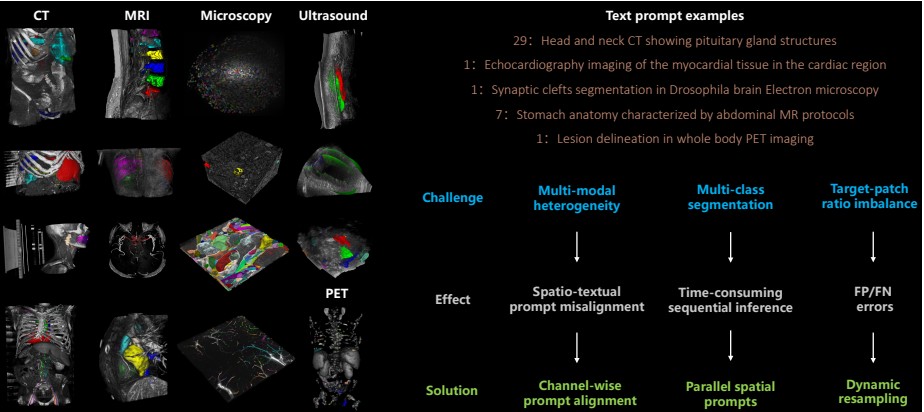

**Fig. 1.** Left: Example renders from the BiomedSegFM challenge dataset (original images and segmentation masks) covering five imaging modalities: CT, MRI, microscopy, PET, and ultrasound. Top-right: Sample text prompts. Bottom-right: Key challenges include (1) multi-modal heterogeneity, (2) multi-class segmentation, and (3) target-patch ratio imbalance, causing spatio-textual misalignment, sequential inference inefficiency, and FP/FN errors. Our solutions: channel-wise prompt alignment (2.3), parallel spatial prompts (2.3), and dynamic resampling (2.5).

## 1   Introduction

Medical image segmentation, the precise delineation of anatomical structures and pathologies within medical volumes, is fundamental to computational healthcare. Despite its importance, challenges persist due to the diversity of imaging modalities and anatomical variations. Recent advances in foundation models, notably the Segment Anything Model (SAM) [13] and its successor, SAM 2 [19], have transformed natural image segmentation by introducing promptable models that generalize across various image distributions and tasks. However, directly applying these models to medical volumes is hindered by the intrinsic differences between natural and medical images.

Adaptations of foundation models for medical image segmentation have followed distinct strategies, each with inherent trade-offs. Early approaches, such as MedSAM [16], extended SAM's 2D capabilities to medical images, primarily using bounding box prompts. Similarly, ScribblePrompt [23], a 2D model, improved segmentation accuracy for unseen labels and image types by supporting flexible annotation styles, including bounding boxes, clicks, and scribbles. To overcome the limitations of 2D methods and leverage 3D spatial information, subsequent models incorporated 3D spatial prompts. For example, SAM-Med3D [22], SegVol [5], and VISTA3D [9] introduced dedicated 3D prompting mechanisms. VISTA3D [9] enabled automatic and interactive 3D segmentation with spatial prompts, facilitating efficient inspection and editing by clinicians. SegVol [5] expanded prompt types to include spatial and semantic cues, im-

proving precision and semantic disambiguation. Beyond this, MedSAM2 [18] advanced 3D segmentation by using intuitive 2D interactions to generate full 3D segmentations. MedSAM2 [18], akin to SAM 2 [19], supports segmentation of 3D medical images and videos, primarily using bounding box prompts and memory-conditioned features.

Most recently, nnInteractive [6], built on the nnU-Net [11] framework, introduced a comprehensive 3D interactive open-set segmentation method. This approach supports diverse spatial prompts, including points, scribbles, boxes, and a novel lasso prompt, leveraging 2D interactions to produce complete 3D segmentations with superior performance. Despite these advancements, current medical segmentation models face significant limitations with spatial prompts. Models like SegVol [5] and SAM-Med3D [22] often rely on multiple downsampling operations for spatial prompts, while VISTA3D [9] downsamples spatial point prompts, leading to substantial loss of voxel-level details. In contrast, nnInteractive [6] incorporates spatial prompts at the native resolution, preserving 3D spatial context. Moreover, existing spatial prompting methods process multiple classes sequentially rather than in parallel, reducing inference efficiency and limiting the model's ability to learn features across interrelated anatomical regions.

On the text prompt front, models like CLIP-Driven Universal Model [15] and SegVol [5] utilize simple semantic classes as prompts, while VISTA3D [9] employs similar class-based prompts. However, such categorical prompting often lacks flexibility in practice. More recent models, such as SAT [28] and BioMedParse [27], adopt text-only prompting paradigms. However, BioMedParse, a 2D model, remains limited in handling 3D medical images. These approaches often sacrifice 3D spatial context and lack spatial prompts, hindering self-iterative refinement and real-world correction capabilities. CAT [10] attempts to integrate spatial anatomical information with text prompts but embeds cropped regions after multiple downsampling steps, failing to utilize spatial prompts at native resolution. Additionally, its complex contrastive learning approach for inter-class relationships is less streamlined. This fragmentation creates a tension between preserving native-resolution spatial prompts and achieving efficient multimodal processing. Ideally, spatial prompts for different classes and their corresponding text should maintain one-to-one channel-wise correspondence at the native resolution. In text-guided controllable generation, several works have successfully integrated native-resolution segmentation masks with text prompts. Prior works like MakeAScene [7] and SpaText [1] have demonstrated effective fusion of segmentation masks and text for controllable generation. ControlNet [25] further enhances this through spatial conditioning of diffusion models. However, such joint textual and native-resolution spatial prompts approaches remain unexplored for medical image segmentation.

To address these challenges, we introduce Medal S, a medical segmentation foundation model that natively supports both spatial and textual prompts in an end-to-end framework. Medal S aligns with initiatives like ScaleMAI [14] and supports datasets including RadGenome-Chest CT [26] and RadGPT [2].

Our key contributions are:

- A novel channel-wise alignment between volumetric prompts and text embeddings through text embedding transformation and lightweight 3D convolution, addressing spatial prompt-text misalignment and enabling precise simultaneous refinement.
- Parallel spatial prompting at native resolution with simultaneous 3D spatial/textual input for multi-class segmentation. Maintains full fidelity and provides more than $10\times$ faster inference (24 classes) vs. sequential processing.
- Dynamic resampling for target-patch ratio imbalance (building upon SAT [28] and nnU-Net [11]), with optimized text preprocessing, two-stage inference, and post-processing, achieving fast inference, memory efficiency, and excellent performance.
- Comprehensive support for 243 classes across CT, MRI, PET, ultrasound, and microscopy (BiomedSegFM dataset), featuring both text-only self-refinement and hybrid manual annotation modes for enhanced clinical flexibility.

## 2   Method

Our proposed **Medal S** framework presents a novel approach to universal medical image segmentation by synergistically integrating spatial prompts with text-driven feature adaptation. As illustrated in Fig. 2, the framework consists of three key components: (1) An image encoder that extracts multi-scale visual features, (2) A text encoder that processes prompt embeddings, and (3) A query decoder that fuses visual and textual features to produce adapted embeddings. Parallel spatial prompts–whether simulated, predicted, or annotated–are processed at native resolution and aligned with spatio-textual features through channel-wise alignment. The framework supports iterative self-refinement for precise segmentation, offering both robustness and flexibility in medical segmentation.

### 2.1   Prompt Encoder

The prompt encoder comprises two components: foreground spatial prompt encoding and textual prompt encoding, with implementations inspired by nnInteractive [6] and SAT [28] respectively.

**Foreground spatial prompt encoding**  To enhance the model's focus on target foreground regions, we generate a foreground spatial prompt $\mathbf{S}_f \in \mathbb{R}^{1 \times H \times W \times D}$ by aggregating parallel spatial prompts $\mathbf{S}_p \in \mathbb{R}^{N \times H \times W \times D}$ (obtained from either previous predictions or user annotations) through a binary thresholding operation:

$$\mathbf{S}_f = H\left(\sum_{i=1}^{N} \mathbf{S}_p^{(i)}\right) \tag{1}$$

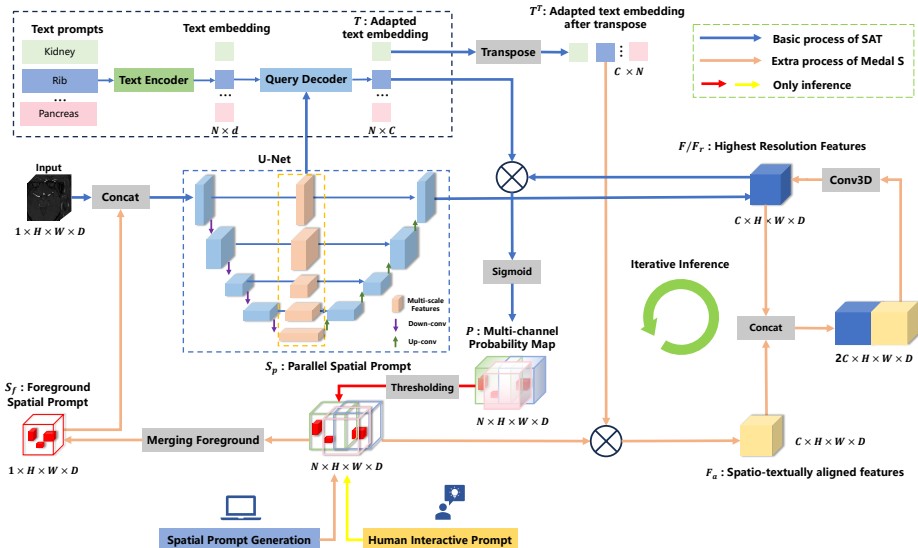

**Fig. 2.** Medal S framework pipeline. Multi-scale visual features from the image encoder and text embeddings from the text encoder are fused by a query decoder into adapted embeddings. Parallel spatial prompts (simulated, predicted, or annotated) are processed at native resolution and aligned via channel-wise matching, maintaining full fidelity. This achieves a greater than $10\times$ speedup for 24-class segmentation versus sequential processing (see Fig. 4) and supports iterative self-refinement for precise segmentation.

where $H(\cdot)$ is the Heaviside step function:

$$H(x) = \begin{cases} 1 & \text{if } x > 0 \\ 0 & \text{otherwise} \end{cases} \tag{2}$$

The resulting $\mathbf{S}_f$ is concatenated with the input image as additional channels to the U-Net encoder, similar to the native resolution prompt in nnInteractive.

**Textual prompt encoding** We employ a frozen pre-trained text encoder $\Phi_{\text{text}}$ from the SAT framework to process medical terminology prompts $\mathcal{T} = \{t_1, \ldots, t_N\}$:

$$\mathbf{z}_j = \Phi_{\text{text}}(t_j), \quad \mathbf{z}_j \in \mathbb{R}^d \tag{3}$$

where $\mathbf{z}_j$ represents the embedding for anatomical target $t_j$.

## 2.2 Spatial Prompt Generation

We introduce a spatial prompt generation method $\mathcal{G}_{\text{prompt}}$ that enhances segmentation robustness for both interactive applications and autonomous refinement.

The generator produces realistic coarse segmentations from ground truth masks $\mathbf{M} \in {0, 1}^{N \times H \times W \times D}$, where $N$ represents semantic channels and $(H, W, D)$ denote spatial dimensions. The method outputs two complementary binary prompts: a single-channel global foreground prompt $\mathbf{S}_f \in {0, 1}^{1 \times H \times W \times D}$ and a multi-channel class-specific prompt $\mathbf{S}_p \in {0, 1}^{N \times H \times W \times D}$.

The generation process applies controlled stochastic transformations through five key parameters. The drop probability range $[p_{\text{drop}}^{\min}, p_{\text{drop}}^{\max}] \in [0, 1]^2$ regulates false negative simulation by removing mask blocks, while the add probability range $[p_{\text{add}}^{\min}, p_{\text{add}}^{\max}]$ controls false positive generation through block additions. Channel-level variations are introduced via $p_{\text{chan-zero}}$, which nullifies entire channels in $\mathbf{S}_p$, and $p$zero determines the probability of returning empty prompts. The block size set $\mathcal{B} \subset \mathbb{Z}_{\geq 1}^3$ defines the possible 3D block dimensions for these transformations.

As detailed in Algorithm 1, the process begins by potentially returning empty prompts when a random sample falls below $p_{\text{zero}}$. Otherwise, $\mathbf{M}_{\text{eff}}$ is created by randomly zeroing out channels in $\mathbf{M}$ according to $p_{\text{chan-zero}}$. The single-channel prompt $\mathbf{S}_f$ is generated by channel summation and binarization of $\mathbf{M}_{\text{eff}}$. For block operations, the method samples block dimensions $[b_h, b_w, b_d] \in \mathcal{B}$, establishing a transformation grid. Mutually exclusive drop and add masks ($\mathbf{B}_{\text{drop}}$ and $\mathbf{B}_{\text{add}}$) are generated using probabilistically sampled parameters, upsampled to full resolution, and applied to both prompt types. The multi-channel prompt additionally incorporates class-specific variations through random channel assignment of modified blocks. Final outputs undergo binarization to maintain strict 0,1 values.

This approach systematically simulates diverse input conditions ranging from coarse segmentations to imperfect user annotations, significantly improving model generalization across varying input qualities while maintaining anatomical plausibility. The stochastic yet controlled transformations enable robust handling of real-world scenarios where prompt quality may vary substantially.

## 2.3   Query Decoder

Our query decoder builds upon the architectures of SAM [13] and SAT [28], while incorporating key design principles from DETR [3] and MaskFormer [4] for segmentation tasks. Departing from conventional approaches that compress dense prompts into low-dimensional mask embeddings (e.g., SAM-Med3D [22], SegVol [5], and VISTA3D [9]), our method introduces a novel preservation of the complete 3D spatial context in volumetric prompts. This preservation proves particularly vital for 3D medical imaging applications, where the maintenance of native spatial resolution directly impacts diagnostic accuracy.

The decoder's architecture establishes precise channel-wise alignment between volumetric prompts and text embeddings, effectively mitigating the accuracy degradation typically caused by resolution mismatches. This design enables efficient parallel processing of multiple native-resolution masks, yielding significant improvements in multi-class segmentation performance. Furthermore, we

---

**Algorithm 1** Spatial Prompt Generation

---

**Require:** $\mathbf{M} \in \{0,1\}^{N \times H \times W \times D}$, $[p_{\text{drop}}^{\min}, p_{\text{drop}}^{\max}]$, $[p_{\text{add}}^{\min}, p_{\text{add}}^{\max}]$, $p_{\text{chan-zero}}$, $p_{\text{zero}} \in [0,1]$,
$\quad \mathcal{B} \subset \mathbb{Z}_{\geq 1}^3$

**Ensure:** $\mathbf{S}_f \in \{0,1\}^{1 \times H \times W \times D}$, $\mathbf{S}_p \in \{0,1\}^{N \times H \times W \times D}$

1: **if** $\text{Random}(0,1) < p_{\text{zero}}$ **then**
2:      **return** $\mathbf{0}_{1 \times H \times W \times D}$, $\mathbf{0}_{N \times H \times W \times D}$
3: **end if**
4:    $\mathbf{M}_{\text{eff}} \leftarrow \mathbf{M} \odot \text{ChannelMask}(\mathbf{M}, p_{\text{chan-zero}})$
5:    $\mathbf{S}_p \leftarrow \mathbf{M}_{\text{eff}}$
6:    $\mathbf{S}_f \leftarrow (\sum_{c=1}^{N} \mathbf{M}_{\text{eff},c} > 0)$
7: **if** $p_{\text{drop}}^{\max} > 0$ or $p_{\text{add}}^{\max} > 0$ **then**
8:      $[b_h, b_w, b_d] \leftarrow \text{RandomChoice}(\mathcal{B})$
9:      $[n_h, n_w, n_d] \leftarrow [\lceil H/b_h \rceil, \lceil W/b_w \rceil, \lceil D/b_d \rceil]$
10:     $p_d \sim \mathcal{U}(p_{\text{drop}}^{\min}, p_{\text{drop}}^{\max})$, $p_a \sim \mathcal{U}(p_{\text{add}}^{\min}, p_{\text{add}}^{\max})$
11:     $\mathbf{B}_{\text{drop}} \leftarrow \text{Random}(n_h, n_w, n_d) < p_d$
12:     $\mathbf{B}_{\text{add}} \leftarrow (\text{Random}(n_h, n_w, n_d) < p_a) \wedge (\neg \mathbf{B}_{\text{drop}})$
13:     $\mathbf{U}_{\text{drop}}, \mathbf{U}_{\text{add}} \leftarrow \text{Upsample}(\mathbf{B}_{\text{drop}}, (H,W,D)), \text{Upsample}(\mathbf{B}_{\text{add}}, (H,W,D))$
14:     $\mathbf{S}_f \leftarrow \mathbf{S}_f \odot (1 - \mathbf{U}_{\text{drop}}) \vee \mathbf{U}_{\text{add}}$
15:     $\mathbf{C}_{\text{keep}} \leftarrow \text{AssignBlocksToChannels}(\neg \mathbf{B}_{\text{drop}}, N)$
16:     $\mathbf{C}_{\text{add}} \leftarrow \text{AssignBlocksToChannels}(\mathbf{B}_{\text{add}}, N)$
17:     $\mathbf{S}_p \leftarrow \mathbf{S}_p \odot \mathbf{C}_{\text{keep}} \vee \mathbf{C}_{\text{add}}$
18: **end if**
19: $\mathbf{S}_f \leftarrow (\mathbf{S}_f > 0)$, $\mathbf{S}_p \leftarrow (\mathbf{S}_p > 0)$
20: **return** $\mathbf{S}_f, \mathbf{S}_p$

---

incorporate a lightweight 3D convolutional module that jointly optimizes voxel-space features using both prompt modalities while maintaining their channel-wise alignment, ensuring accurate and robust 3D segmentation across targets with varying semantics.

The query decoder operates on per-voxel features $\mathbf{F} \in \mathbb{R}^{C \times H \times W \times D}$, where $(H, W, D)$ denote the voxel grid dimensions and $C$ represents the feature channels. These features are derived through progressive upsampling of visual encoder outputs with skip connections in a U-Net-style architecture [20]. Concurrently, the decoder receives adapted text embeddings $\mathbf{T} \in \mathbb{R}^{N \times C}$ produced by a transformer-based query decoder [21], where $N$ indicates the number of semantic queries (corresponding to anatomical targets). The query decoder adapts text embeddings $\mathbf{Z} \in \mathbb{R}^{N \times L}$ (with $L$ as the text dimension) using multi-scale visual features $\mathbf{V} \in \mathbb{R}^{C_V \times H_V \times W_V \times D_V}$ according to:

$$\mathbf{T} = \Phi_{\text{query}}(\mathbf{V}, \mathbf{Z})$$

The core innovation of our approach lies in the spatial prompt refinement module. This module enhances per-voxel features $\mathbf{F}$ through an interaction mechanism between adapted text embeddings $\mathbf{T}$ and parallel spatial prompts $\mathbf{S}_p \in \mathbb{R}^{N \times H \times W \times D}$ (which originate from either previous predictions or user an-

notations during inference). The refinement process begins with the computation of spatio-textually aligned features $\mathbf{F}_a \in \mathbb{R}^{C \times H \times W \times D}$ via:

$$\mathbf{F}_a = \mathbf{T}^\top \mathbf{S}_p$$

where $\mathbf{T}^\top \in \mathbb{R}^{C \times N}$ denotes the transposed adapted text embeddings. We then concatenate $\mathbf{F}_a$ with the original features $\mathbf{F}$ along the channel dimension, resulting in a $\mathbb{R}^{2C \times H \times W \times D}$ tensor. This combined representation is processed by a lightweight 3D convolutional module inspired by nnU-Net's native-resolution skip connection architecture [11], producing refined features $\mathbf{F}_r \in \mathbb{R}^{C \times H \times W \times D}$:

$$\mathbf{F}_r = \text{Conv}([\mathbf{F}; \mathbf{F}_a])$$

The final per-voxel prediction $\mathbf{P} \in \mathbb{R}^{N \times H \times W \times D}$ is obtained through voxel-wise correlation between queries and refined features, followed by sigmoid activation $\sigma(\cdot)$:

$$\mathbf{P} = \sigma\left(\mathbf{T}\mathbf{F}_r\right)$$

This approach produces a multi-channel probability map with dedicated channels for each anatomical structures and pathological region, facilitating complete 3D volumetric segmentation.

### 2.4   Iterative Inference

Our query decoder employs an iterative inference approach inspired by Masked Autoencoders (MAE) [8]. The algorithm progressively refines predictions through multiple iterations, where each output $\mathbf{P}^{(t)} \in \mathbb{R}^{N \times H \times W \times D}$ serves as the spatial prompt $\mathbf{S}_p$ for subsequent iterations. The random masking mechanism facilitates prediction in unprompted regions, with complementary masked predictions aggregated to improve robustness against input noise.

As detailed in Algorithm 2, each iteration $t$ consists of four key components: (1) feature enhancement through cross-attention between text queries $\mathbf{T}$ and spatial prompts, (2) $R$ rounds of random block masking using block sizes $\mathcal{B} = 4, 8$, (3) parallel prediction of both masked and unmasked regions, and (4) prediction averaging across all rounds.

Medal-S supports two distinct prompting strategies. The text-only mode initializes with zero tensors and relies solely on text prompts with self-refinement, while the hybrid mode incorporates external spatial cues such as manual annotations when configured. The inference pipeline orchestrates prompt generation and iterative refinement through coordinated function calls, dynamically updating spatial prompts to enhance segmentation accuracy throughout the process.

### 2.5   Dynamic Resampling

Dynamic resampling addresses the challenge of varying segmentation target sizes relative to a fixed patch size in medical image segmentation. When the target

---

**Algorithm 2** Iterative Query Decoder Inference

---

**Require:** Voxel features $\mathbf{F} \in \mathbb{R}^{C \times H \times W \times D}$, queries $\mathbf{T} \in \mathbb{R}^{N \times C}$, initial prompt $\mathbf{S}_p \in \mathbb{R}^{N \times H \times W \times D}$, iterations $T$, parameters $\Theta$, block sizes $\mathcal{B} = \{4, 8\}$, repetitions $R = 1$

**Ensure:** Prediction $\mathbf{P} \in \mathbb{R}^{N \times H \times W \times D}$

1:   $\mathbf{P}^{(0)} \leftarrow \mathbf{0}$
2:   **for** $t = 1$ to $T$ **do**
3:      $\mathbf{E}^{(t)} \leftarrow \mathbf{T}^{\top} \mathbf{S}_p^{(t-1)}$
4:      $\mathbf{F}_r^{(t)} \leftarrow \text{Conv}([\mathbf{F}; \mathbf{E}^{(t)}]; \Theta)$
5:      $\mathbf{P}^{(t)} \leftarrow \sigma(\mathbf{T}\mathbf{F}_r^{(t)})$
6:      $\mathbf{S}_p^{(t)} \leftarrow \mathbf{P}^{(t)}$
7:      $\mathbf{P}_{\text{sum}} \leftarrow \mathbf{0}$
8:      **for** $r = 1$ to $R$ **do**
9:         $b \leftarrow \text{RandomChoice}(\mathcal{B})$
10:       $N_h \leftarrow \lceil H/b \rceil$, $N_w \leftarrow \lceil W/b \rceil$, $N_d \leftarrow \lceil D/b \rceil$
11:       $N_{\text{selected}} \leftarrow \max(1, \lfloor (N_h \cdot N_w \cdot N_d)/2 \rfloor)$
12:       $\mathbf{M}_b \leftarrow \text{RandomMask}(N_h, N_w, N_d, N_{\text{selected}})$
13:       $\mathbf{M} \leftarrow \text{Upsample}(\mathbf{M}_b, (H, W, D))$
14:       $\mathbf{M}_c \leftarrow 1 - \mathbf{M}$
15:       $\mathbf{P}_1 \leftarrow \text{Model}(\mathbf{T}, \mathbf{P}^{(t)}, \mathbf{M}; \Theta)$
16:       $\mathbf{P}_2 \leftarrow \text{Model}(\mathbf{T}, \mathbf{P}^{(t)}, \mathbf{M}_c; \Theta)$
17:       $\mathbf{P}_{\text{patch}} \leftarrow \mathbf{P}_1 \cdot \mathbf{M}_c + \mathbf{P}_2 \cdot \mathbf{M}$
18:       $\mathbf{P}_{\text{sum}} \leftarrow \mathbf{P}_{\text{sum}} + \mathbf{P}_{\text{patch}}$
19:      **end for**
20:      $\mathbf{P}^{(t)} \leftarrow \mathbf{P}_{\text{sum}}/R$
21: **end for**
22: **return** $\mathbf{P}^{(T)}$

---

size significantly exceeds the patch size, partial visibility of the target within a patch can lead to false positives (FP), as the model lacks global context and may misinterpret background noise as part of the target. Conversely, when the target is much smaller than the patch, the imbalance between foreground and background can result in false negatives (FN), as the model struggles to focus on small, critical regions. To mitigate these issues, we propose a dynamic resampling strategy that adjusts the voxel spacing of the input image based on the physical size of the smallest foreground connected component or the smallest class-specific target, ensuring balanced representation within the fixed patch size used by the model.

Our approach begins by identifying the smallest foreground connected component or the smallest target class in each case, which serves as the reference for resampling. Ideally, each target would have a tailored resampling rate to optimize its representation within the patch. However, to balance computational efficiency during training and inference, we focus on the smallest target to determine the resampling parameters. The core idea is to adjust the current spacing $\mathbf{s} = [s_x, s_y, s_z]$ of the image to a target spacing $\mathbf{t} = [t_x, t_y, t_z]$ such that the physical dimensions of the target align with the patch size $\mathbf{p} = [p_x, p_y, p_z]$. The adjusted spacing for each dimension $i$ is computed as:

$$s_i' = \begin{cases} \max\left(t_i, \frac{p_i \cdot \alpha \cdot t_i}{d_i}\right), & \text{if } s_i > t_i, \\ \min\left(t_i, \frac{p_i \cdot \alpha \cdot t_i}{d_i}\right), & \text{otherwise,} \end{cases}$$

where $s_i'$ is the adjusted spacing, $d_i$ is the image dimension, and $\alpha$ is a scale factor. This formula ensures that the physical size of the resampled image fits within the patch while preserving sufficient detail. To prevent excessive resampling, we impose constraints such that $s_i'$ remains within practical bounds, depending on the target class and inference stage.

## 2.6   Two-stage Inference

We propose a two-stage inference strategy to optimize computational efficiency and segmentation accuracy for medical imaging, particularly for localized regions like focal lesions or anatomical structures. The coarse-to-fine strategy first performs low-resolution segmentation to identify regions of interest (ROIs), followed by high-resolution refinement to capture fine details. This approach is efficient for datasets with small foreground regions, reducing inference time compared to full high-resolution processing.

The strategy trains two models: one for coarse segmentation at a voxel spacing of (1.5, 1.5, 3.0) and another for high-resolution segmentation at (1.0, 1.0, 1.0). In the first stage, images are processed using a sliding window with a crop size of (224, 224, 128), corresponding to a physical field of view of approximately (336, 336, 384). This coarse resolution enables rapid ROI detection. The output mask highlights potential target regions, but if no foreground is detected, the strategy defaults to full-volume high-resolution inference to avoid missing subtle targets.

In the second stage, the ROI is extracted based on the coarse segmentation's non-zero predictions, scaled by a factor (1.1 to 1.5) to include context. The image is resampled to a target spacing of (1.0, 1.0, 1.0) with a crop size of (192, 192, 192). To manage memory, the physical volume $V = \prod_{i=1}^{3} s_i \cdot d_i$ (where $s_i$ and $d_i$ are voxel spacing and dimension size) is constrained by a threshold $V_{\text{threshold}} = (1.8)^3 \cdot \prod_{i=1}^{3} c_i$, with $c_i$ as the crop size. If exceeded, voxel spacing is adjusted to satisfy $s_i \cdot d_i \leq 1.9 \cdot c_i \cdot t_i$, where $t_i$ is the target spacing, ensuring memory usage stays below 32 GB.

The second stage refines segmentation using the coarse predictions as spatial prompts, enhancing accuracy for small or intricate structures like synaptic clefts or micro-lesions. A sliding window approach ensures high precision despite increased computational cost. The strategy allows flexible use of either stage: the first for rapid analysis or the second for high-resolution segmentation when resources permit. This adaptability balances efficiency and accuracy, making the approach suitable for diverse medical imaging applications.

## 2.7 Text Prompts Preprocessing

To effectively preprocess text prompts from the BiomedSegFM dataset, a systematic approach is employed to extract modality and class-specific identifiers, enabling dynamic resampling and post-processing strategies. The methodology begins by parsing the text prompts JSON to generate a class mapping, which assigns unique identifiers to anatomical structures and lesions across modalities (CT, MRI, US, PET, Microscopy). This mapping is constructed by extracting modality information from dataset prefixes and standardizing class names, such as mapping "Left renal structure" to "Left kidney" or "Myocardium" to "Heart". The resulting class mapping, stored as a JSON file, ensures each class within a modality has a unique identifier, facilitating consistent encoding.

Next, a variant mapping is created to handle diverse terminologies in the prompts. This mapping accounts for anatomical and lesion variants, incorporating directionality (e.g., "left" or "right") and suffixes (e.g., "lesions", "tumors"). For instance, "hepatic lesions" is mapped to "Liver lesions" using predefined rules and regular expressions to detect directional patterns. The variant mapping prioritizes longer, more specific terms to avoid partial matches, ensuring "Brainstem" is distinguished from "Brain". This preprocess yields a comprehensive variant mapping JSON, covering all prompt variations.

For training and inference, a text preprocessing function extracts modality and class information from each prompt. Given a sentence $s$ and instance label $l$ (0 for anatomy, 1 for lesion), the function identifies the modality $m$ (e.g., CT, MRI, US, PET, or Microscopy) by matching keywords. It then retrieves the class identifier $c_{id}$ and canonical name $c_{name}$ from the class mapping, using the variant mapping to handle term variations. The function prioritizes longer matches to ensure specificity, formalized as:

$$(c_{id}, c_{name}) = \arg \max_{k \in K} \left( \text{len}(k) \mid k \in s, k \in M_m^l \right),$$

where $K$ is the set of terms (class names and variants), $M_m^l$ is the modality-specific class dictionary for label $l$, and $\text{len}(k)$ is the term length. Directional patterns are detected using regular expressions to refine matches, such as distinguishing "Left kidney" from "Kidney".

The extracted modality $m$ and class identifier $c_{id}$ are input to a text encoder, producing embeddings that guide dynamic resampling and post-processing. For example, the extracted modality $m$ enables the text encoder to distinguish between different modalities, while resampling strategies leverage $c_{id}$ to apply class-specific target spacing, or post-processing leverages $c_{id}$ to apply class-specific segmentation refinements. This streamlined pipeline ensures robust handling of diverse text prompts, enhancing segmentation accuracy.

## 2.8 Loss Function

We employ a combined loss $\mathcal{L} = \mathcal{L}_{BCE} + \mathcal{L}_{Dice}$, standard in medical image segmentation. For $N$ classes and $C$ voxels:

$$\mathcal{L}_{BCE} = -\frac{1}{MC} \sum n, c \left[ s_{n,c} \log p_{n,c} + (1 - s_{n,c}) \log(1 - p_{n,c}) \right]$$
$$\mathcal{L}_{Dice} = 1 - \frac{2 \sum n, c p_{n,c} s_{n,c}}{\sum_{n,c} p_{n,c}^2 + \sum_{n,c} s_{n,c}^2}$$

where $p_{n,c}$ and $s_{n,c}$ are the predicted probability and ground truth (0 or 1) for class $n$ at voxel $c$. This combination optimizes both pixel-level accuracy and region-based overlap.

### 2.9   Post-processing

Our post-processing method refines segmentation results by suppressing spurious predictions while preserving anatomically plausible structures, improving upon nnU-Net [11]. Unlike nnU-Net, which retains only the largest connected component across all classes in a single operation, our approach processes each class independently and leverages probability maps to prioritize components based on both probability and size. As outlined in Algorithm 3, given a probability map $\mathbf{P} \in \mathbb{R}^{N \times H \times W \times D}$, where $N$ is the number of classes, the segmentation map $\mathbf{S} \in \mathbb{R}^{1 \times H \times W \times D}$ is derived by computing the maximum probability across classes, $p_{\max} = \max_{j=1,\ldots,N} \mathbf{P}_j$, and the corresponding class index, $c_{\max} = \arg\max_{j=1,\ldots,N} \mathbf{P}_j$. Voxels are assigned class labels $l_j \in \{1, \ldots, N\}$ where $p_{\max} \geq 0.5$, i.e., $\mathbf{S} = l_{c_{\max}}$ if $p_{\max} \geq 0.5$, otherwise $\mathbf{S} = 0$ (background).

For each class $l \in \{1, \ldots, N\}$, a binary mask $M_l = (\mathbf{S} = l)$ is created. Connected components in $M_l$ are labeled using 6-connectivity, yielding a labeled image $C_l$ and component sizes $\Sigma_l = \{(c_i, s_i)\}$. The mean probability for component $c_i$ is computed as the average of $\mathbf{P}_l$ over voxels where $C_l = c_i$. Among the top three largest components, those with mean probabilities within $\tau = 0.1$ of the maximum and above 0.86 are retained. If none qualify, the highest-probability component is kept if it is among the two largest and its size is at least 0.6 times the largest; otherwise, the largest component is selected. The refined mask $M_l'$ updates $\mathbf{S}$ by setting $\mathbf{S} = 0$ where $M_l \land \neg M_l'$. This method enhances nnU-Net by processing classes individually and using probabilities to guide component selection, improving multi-class segmentation robustness.

## 3   Experimental Setup

### 3.1   Data and Evaluation Methodology

The development set builds upon the CVPR 2024 MedSAM on Laptop Challenge [17], incorporating additional 3D cases sourced from publicly available datasets[5]. This collection encompasses various standard 3D imaging modalities,

---

[5] Complete dataset details can be found at https://medsam-datasetlist.github.io/

---

**Algorithm 3** Post-processing

---

**Require:** $\mathbf{P} \in \mathbb{R}^{N \times H \times W \times D}$, labels $\{1, \ldots, N\}$, background $b = 0$, threshold $\tau = 0.1$,
        connectivity $k = 6$
**Ensure:** Refined segmentation $\mathbf{S} \in \mathbb{R}^{1 \times H \times W \times D}$
  1: $p_{\max} \leftarrow \max_{j=1,\ldots,N} \mathbf{P}_j$, $c_{\max} \leftarrow \arg\max_{j=1,\ldots,N} \mathbf{P}_j$
  2: $\mathbf{S} \leftarrow 0$, $\mathbf{S}[p_{\max} \geq 0.5] \leftarrow l_{c_{\max}}$
  3: **for** $l = 1$ to $N$ **do**
  4:       $M_l \leftarrow (\mathbf{S} = l)$
  5:       $C_l, \Sigma_l \leftarrow \text{ConnectedComponents}(M_l, k)$
  6:       **if** $\Sigma_l = \emptyset$ **then continue**
  7:       **end if**
  8:       $T \leftarrow \text{SortBySize}(\Sigma_l)[: 3]$
  9:       $P_T \leftarrow \{(c_i, \text{mean}(\mathbf{P}_l[C_l = c_i])) \mid c_i \in T\}$
 10:       $p_{\max} \leftarrow \max\{p_i \mid (c_i, p_i) \in P_T\}$
 11:       $K \leftarrow \{c_i \mid (c_i, p_i) \in P_T, (p_{\max} - p_i) \leq \tau, p_i > 0.86\}$
 12:       **if** $|K| \geq 2$ **then**
 13:           $M_l' \leftarrow (C_l \in K)$
 14:       **else**
 15:           $c_{\max} \leftarrow \arg\max_{c_i}\{p_i \mid (c_i, p_i) \in P_T\}$
 16:           $T_2 \leftarrow T[: 2]$
 17:           **if** $c_{\max} \in T_2$ and $\Sigma_l(c_{\max})/\Sigma_l(T[0]) > 0.6$ **then**
 18:               $M_l' \leftarrow (C_l = c_{\max})$
 19:           **else**
 20:               $M_l' \leftarrow (C_l = T[0])$
 21:           **end if**
 22:       **end if**
 23:       $\mathbf{S}[M_l \wedge \neg M_l'] \leftarrow b$
 24: **end for**
 25: **return S**

---

including Computed Tomography (CT), Magnetic Resonance Imaging (MRI), Positron Emission Tomography (PET), Ultrasound, and Microscopy. The hidden test set was collaboratively developed by the community, consisting exclusively of previously unpublished cases. All annotations were either supplied by data contributors or generated by the challenge organizers using 3D Slicer [12] and MedSAM2 [18]. Participants have the option to either use the full training set or participate in the coreset track, which permits model development using only 10% of the total training cases.

The text-guided segmentation task evaluates both semantic and instance segmentation performance. Semantic segmentation assessment employs two metrics: the Dice Similarity Coefficient (DSC) for measuring region overlap and Normalized Surface Distance (NSD) for evaluating boundary accuracy. Instance segmentation performance is quantified using the F1 score at a 0.5 overlap threshold, along with DSC scores for correctly identified instances. A runtime constraint of 60 seconds per class is enforced - submissions exceeding this limit will receive zero scores for all DSC and NSD metrics on the affected test cases.

### 3.2   Implementation Details

**Data preprocessing** Consistent with MedSAM [16], all medical images were converted to npz format and normalized to an intensity range of $[0, 255]$. For CT scans specifically, we performed Hounsfield unit normalization using standard window settings: soft tissues (width:400, level:40), lung (width:1500, level:-160), brain (width:80, level:40), and bone (width:1800, level:400). Following this normalization, the intensity values were linearly scaled to the target range of $[0, 255]$. For non-CT imaging modalities, we first truncated intensity values at the 0.5th and 99.5th percentiles before applying the same rescaling procedure. Images that already had native intensity values within the $[0, 255]$ range underwent no additional preprocessing steps.

**Table 1.** Training protocols.

| Pre-trained Model | SAT |
|---|---|
| Batch size | 4 |
| Patch size | 224×224×128 |
| Spacing | (1.5, 1.5, 3.0) |
| Total steps | 108600 |
| Optimizer | AdamW |
| Initial learning rate (lr) | 1e-4 |
| Lr decay schedule | cosine |
| Training time | 168 hours |
| Loss function | BCE+Dice |
| Number of model parameters | 221M |

**Table 2.** Training protocols for the 2nd model

| Pre-trained Model | SAT |
|---|---|
| Batch size | 8 |
| Patch size | 192×192×192 |
| Spacing | (1.0, 1.0, 1.0) |
| Total steps | 91300 |
| Optimizer | AdamW |
| Initial learning rate (lr) | 1e-4 |
| Lr decay schedule | cosine |
| Training time | 160 hours |
| Loss function | BCE+Dice |
| Number of model parameters | 221M |

**Training Protocols** To handle large-scale datasets for fast preprocessing and data loading, the dynamic resampling strategy mentioned in Section 2.5 is de-

veloped based on the latest version of the resampling function from the nnU-Net [11] framework. The resampling function in the latest version of nnU-Net significantly improves the efficiency of resampling large-scale data by leveraging CPU processing. Additionally, the latest Blosc2 compression format from nnU-Net is adopted to compress npz files, achieving a balance between file storage size and read speed during dataloader operations. For the preprocessing of 3D images, dataloader, and data augmentation, most of the functions and code from nnU-Net are retained with appropriate modifications.

For the text dataloader and simultaneous training across multiple datasets, different sampling rates are set for each dataset, along with adjustments to the positive-negative sample ratio and padding alignment for varying batch text prompt lengths. Multi-GPU training is primarily based on the SAT [28]. The optimal model selection criteria are also based on SAT, as the learning rate curve decreases gradually with each epoch; by default, the model from the final iteration is adopted.

The training configurations are as follows: for the 224×224×128 input size, we use a batch size of 2 per GPU across 2 GPUs (effective batch size of 4), while for the 192×192×192 input size, we employ a batch size of 2 per GPU across 4 GPUs (effective batch size of 8). The complete training process takes 7 days (168 hours) to complete. Detailed environment settings and training protocols are presented in Table 3,Table 1, and Table 2.

**Environment settings** The development environments and requirements are presented in Table 3.

**Table 3.** Development environments and requirements.

| System | Ubuntu 22.04.4 LTS (Jammy Jellyfish) |
|---|---|
| CPU | Intel(R) Xeon(R) Platinum 8468 CPU @2.10GHz |
| RAM | 2TB DDR (1.8TB available) |
| GPU (number and type) | Eight NVIDIA H100 80GB HBM3 |
| CUDA version | 12.2 |
| Programming language | Python 3.10.16 |
| Deep learning framework | torch 2.2.0, torchvision 0.17.0 |

## 4    Results and discussion

### 4.1    Quantitative Results on Validation Set

Table 4 presents validation results on the all-data track. Our Medal S model consistently outperforms the CAT and SAT baselines, as well as the single-stage ablation Medal S (w/o stage1), across nearly all metrics and modalities. It falls

**Table 4.** Quantitative evaluation results of the validation set on the **all-data track**.

| Modality | Method | Semantic Segmentation | | Instance Segmentation | |
|---|---|---|---|---|---|
| | | DSC | NSD | F1 | DSC TP |
| CT | CAT | 72.11 | 72.27 | 29.93 | 37.17 |
| | BiomedParse-V | 85.12 | 89.65 | 51.19 | 67.49 |
| | SAT | 67.80 | 67.26 | 25.17 | 39.54 |
| | Medal S (w/o stage1) | 80.36 | 79.28 | 29.66 | 47.14 |
| | Medal S (w/o post) | 81.87 | 81.59 | 38.46 | 49.97 |
| | Medal S | 81.90 | 81.61 | 39.97 | 50.94 |
| MRI | CAT | 54.15 | 61.93 | 13.75 | 28.13 |
| | BiomedParse-V | 73.96 | 86.64 | 53.17 | 70.53 |
| | SAT | 56.10 | 66.69 | 12.28 | 27.28 |
| | Medal S (w/o stage1) | 62.90 | 71.11 | 33.62 | 65.11 |
| | Medal S (w/o post) | 61.96 | 71.11 | 47.50 | 66.48 |
| | Medal S | 61.95 | 70.94 | 46.99 | 66.41 |
| Microscopy | CAT | - | - | 3.13 | 36.28 |
| | BiomedParse-V | - | - | 19.39 | 65.52 |
| | SAT | - | - | 20.06 | 42.43 |
| | Medal S (w/o stage1) | - | - | 33.03 | 70.49 |
| | Medal S (w/o post) | - | - | 33.82 | 72.12 |
| | Medal S | - | - | 33.44 | 72.39 |
| PET | CAT | - | - | 10.98 | 27.79 |
| | BiomedParse-V | - | - | 31.32 | 71.85 |
| | SAT | - | - | 42.00 | 78.63 |
| | Medal S (w/o stage1) | - | - | 28.30 | 72.92 |
| | Medal S (w/o post) | - | - | 33.89 | 70.89 |
| | Medal S | - | - | 32.57 | 72.11 |
| Ultrasound | CAT | 85.94 | 83.60 | - | - |
| | BiomedParse-V | 90.50 | 91.35 | - | - |
| | SAT | 85.58 | 79.24 | - | - |
| | Medal S (w/o stage1) | 77.99 | 73.04 | - | - |
| | Medal S (w/o post) | 82.37 | 79.16 | - | - |
| | Medal S | 82.45 | 79.48 | - | - |
| Average | CAT | 70.73 | 72.60 | 14.45 | 32.34 |
| | BiomedParse-V | 83.19 | 89.21 | 38.77 | 68.85 |
| | SAT | 69.83 | 71.06 | 24.88 | 46.97 |
| | Medal S (w/o stage1) | 73.75 | 74.48 | 31.15 | 63.91 |
| | Medal S (w/o post) | 75.40 | 77.28 | 38.42 | 64.87 |
| | Medal S | 75.44 | 77.34 | 38.24 | 65.46 |

below the concurrent BiomedParse-V in semantic segmentation but achieves comparable performance in instance segmentation.

On average, Medal S achieves 75.44 DSC, 77.34 NSD, 38.24 F1, and 65.46 DSC TP—representing gains of 4.71 / 4.74 / 23.79 / 33.12 over CAT and 5.61 / 6.28 / 13.36 / 18.49 over SAT. These improvements arise from two core innovations. Medal S advances beyond SAT through channel-wise spatial-textual alignment using lightweight 3D convolutions, preserving full-resolution fidelity. Combined with parallel prompt processing and optimized resampling, this enables superior boundary delineation and instance separation.

In ultrasound, Medal S lags behind SAT (82.45 vs. 85.58 DSC) due to large target-to-patch ratios, revealing that the current dynamic resampling strategy still has room for improvement to better adapt to more complex input sizes, spacings, and target proportions. While Medal S demonstrates the above advantages, it still lags behind BiomedParse-V in certain aspects, indicating room for further improvement in future work.

**Ablation on two-stage inference.** The two-stage inference paradigm: low-resolution predictions from stage 1 provide global spatial prompts that effectively guide fine-grained refinement in stage 2. Ablation studies reveal average DSC gains of 1.69, NSD gains of 2.86, F1 gains of 7.09 and DSC TP gains of 1.55, with up to 13.37 F1 improvement in MRI—demonstrating that coarse-to-fine spatial guidance is essential for precise localization in complex, multi-organ scenes.

**Ablation on post-processing.** Medal S with post-processing yields only marginal average gains over Medal S (w/o post): +0.04 DSC, +0.06 NSD, -0.18 F1, +0.59 DSC TP. This suggests that the current post-processing is not universally effective and may disrupt refined predictions in some cases.

In summary, Medal S establishes a new standard in universal medical segmentation by synergistically integrating global spatial cues with precise textual alignment in a two-stage, full-resolution framework—delivering superior accuracy and efficiency across diverse anatomies and imaging modalities.

## 4.2 Qualitative results on validation set

As illustrated in Fig. 3, the qualitative results on the validation set provide insights into the performance of our proposed method, Medal S, across different modalities. Due to the unavailability of the CAT all-data Docker and the lack of updated qualitative results for CAT, a direct comparison with CAT on the all-data track for the validation set is not feasible. Consequently, our analysis focuses on comparing Medal S predictions against ground truth annotations across five modalities, presenting both successful and challenging segmentation cases for each.

The qualitative results reveal that Medal S performs effectively in segmenting multi-class targets and regions with substantial volume across various modalities. In such cases, the model accurately delineates boundaries and captures structural details, benefiting from its channel-wise alignment of volumetric prompts and text embeddings, as well as its ability to process spatial prompts at native

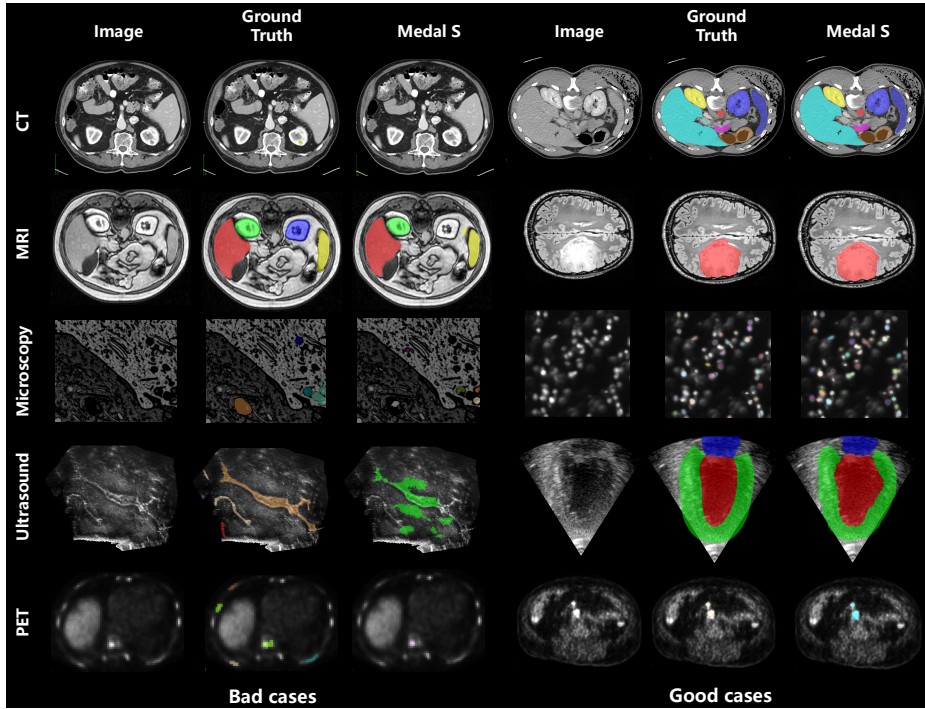

**Fig. 3.** Comparison of Medal S and ground truth results on the validation set for five different modalities. For each modality, we present both good segmentation results and bad segmentation results.

resolution. However, segmentation quality diminishes for smaller lesions, particularly in datasets with significant foreground-background imbalance or blurred boundaries, such as those involving tumors. These challenging cases often exhibit ambiguous edges and complex textures, which pose difficulties for precise segmentation.

A contributing factor to these failures is the inherent noise in the labels of such data, which increases segmentation difficulty. Small lesions and imbalanced datasets amplify the impact of label inaccuracies, making it harder for the model to distinguish between foreground and background. While Medal S's dynamic resampling and optimized preprocessing mitigate some of these issues, further improvements in handling noisy labels and refining boundary detection for small, ambiguous targets are necessary to enhance performance in these scenarios.

### 4.3   Spatial Prompt Ablation Study

As shown in Fig. 5, we conduct a case study to analyze the impact of spatial prompts under three distinct settings: 1) without any spatial prompts, 2) using the Stage-1 segmentation prediction as the spatial prompt for Stage-2, and 3) us-

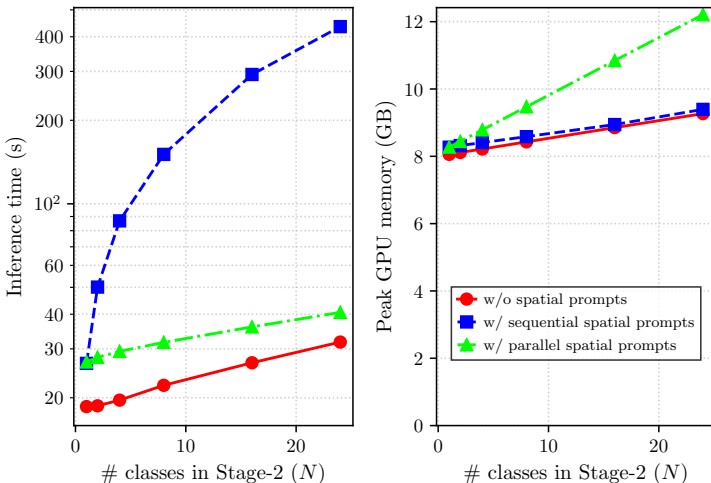

**Fig. 4.** Efficiency comparison of spatial prompting strategies. (a) Inference runtime and (b) peak GPU memory consumption versus the number of classes. Parallel prompting achieves minimal time complexity with respect to the number of classes, resulting in a greater than $10\times$ speedup for 24-class segmentation over the sequential approach, whose runtime grows substantially. While parallel prompting requires moderately more memory, it remains within practical limits and offers a favorable trade-off for drastic time savings in multi-class scenarios.

**Table 5.** Quantitative ablation study on the effect of spatial prompts.

| Variant | DSC | NSD |
|---|---|---|
| w/o spatial prompts | 83.50 | 78.15 |
| w/ stage1 spatial prompts | 83.98 | 79.16 |
| w/ GT spatial prompts | 87.23 | 84.50 |

ing the ground-truth (GT) segmentation mask as the spatial prompt for Stage-2. The results qualitatively demonstrate that the model can dynamically adapt its predictions based on the presence and quality of spatial prompts. Specifically, when provided with more accurate spatial prompts (e.g., GT masks), the model generates significantly refined outputs—effectively suppressing small-scale noise, resolving category confusion, and enhancing spatial continuity. These observations are quantitatively supported by the results in Table 5, where using GT spatial prompts yields the highest DSC and NSD, confirming that the model effectively leverages superior spatial prompts to produce more accurate segmentation results.

Based on the comprehensive analysis of spatial prompt effectiveness in Fig. 5 and Table 5, which is an analysis result for a single test data example, we further demonstrate the efficiency of our proposed parallel spatial prompting mechanism through runtime and memory evaluation. As shown in Fig. 4, our parallel spa-

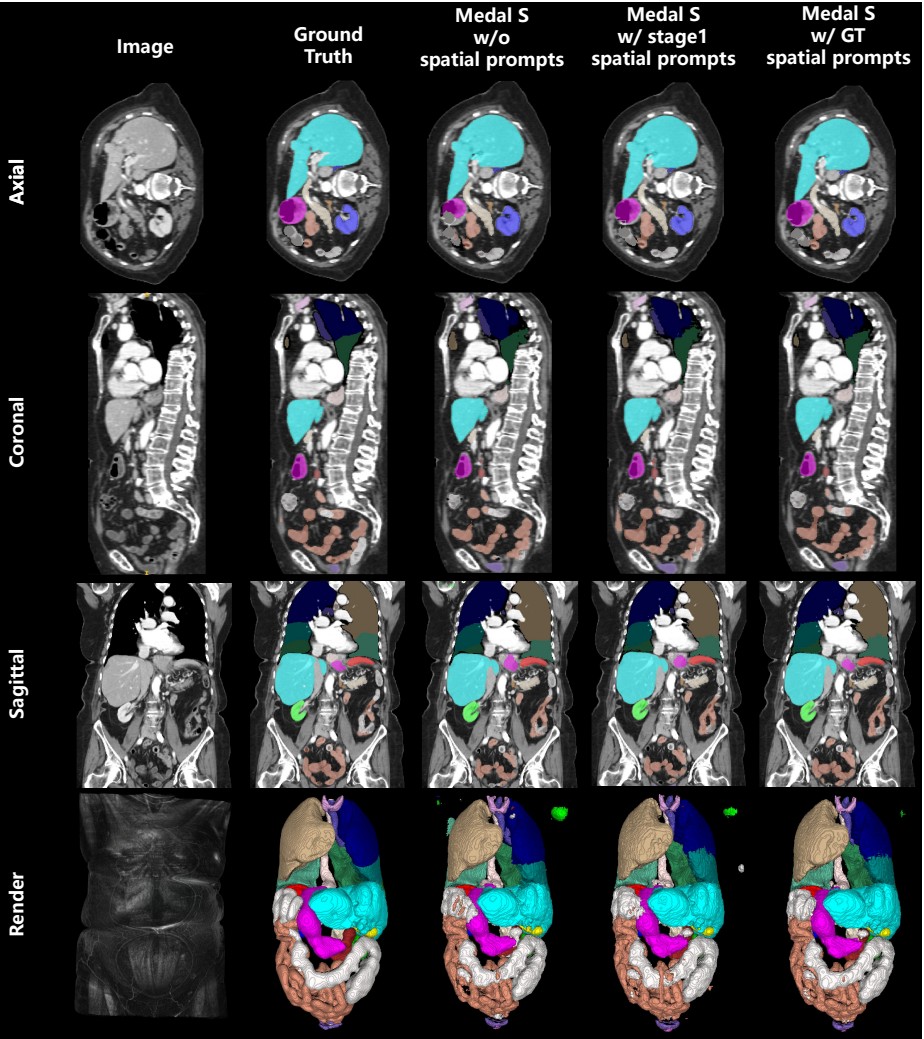

**Fig. 5.** Qualitative comparison of Medal S with different spatial prompt configurations. From left to right: (a) Input Image, (b) Ground Truth, (c) Medal S without spatial prompts, (d) Medal S with Stage-1 prediction as spatial prompts, and (e) Medal S with GT masks as spatial prompts. Each configuration is visualized in axial, coronal, sagittal views and 3D rendering, demonstrating the progressive improvement in segmentation quality with better spatial prompts - particularly in noise reduction, confusion resolution, and continuity enhancement.

tial prompting approach achieves a remarkable balance between performance and efficiency when evaluated on the same single case. While providing significant segmentation quality improvements as evidenced by the 83.98 DSC and 79.16 NSD in Table 5, the parallel method only introduces moderate computational overhead compared to the no-prompt baseline. Crucially, as the number of classes increases from 1 to 24, the parallel approach maintains reasonable runtime (40.63s vs 31.75s baseline) and memory usage (12.5GB vs 9.49GB baseline), while sequential prompting exhibits substantial time growth (435.1s), becoming practically infeasible for multi-class segmentation tasks. This efficiency advantage, combined with the proven effectiveness in enhancing segmentation quality, makes our parallel spatial prompting strategy highly promising for clinical applications where both accuracy and computational efficiency are paramount.

### 4.4   Results on final testing set

The quantitative evaluation results on the test set for the all-data track, as shown in Table 6, demonstrate that our method Medal S achieves a Dice score of 58.06 and an NSD of 59.11, both of which are higher than those of the improved baseline model SAT (DSC: 54.13, NSD: 52.97). This indicates that Medal S is a promising approach. However, there remains a performance gap compared to the current leading method, BiomedParse-V (DSC: 74.97, NSD: 77.47), suggesting potential for further optimization in the future.

**Table 6.** Quantitative evaluation results of the testing set on the **all-data track**.

| Method | Semantic Segmentation | | Instance Segmentation | |
|---|---|---|---|---|
| | DSC | NSD | F1 | DSC TP |
| CAT | 33.04 | 31.53 | 1.94 | 4.66 |
| BiomedParse-V | 74.97 | 77.47 | 23.80 | 44.76 |
| SAT | 54.13 | 52.97 | 14.19 | 29.59 |
| Medal S | 58.06 | 59.11 | 10.54 | 17.86 |

### 4.5   Limitation and future work

While Medal S demonstrates strong performance across multiple modalities, certain limitations warrant further exploration. The dynamic resampling strategy, although effective in addressing target-patch ratio imbalances, requires additional refinement to better handle modalities like ultrasound, where large target sizes challenge the current patch-based approach. Expanding the variety of spatial prompts could further enhance performance. For instance, incorporating diverse spatial prompts, such as those in nnInteractive, including 2D spatial cues, could improve flexibility and precision in capturing complex anatomical structures.

Future work will focus on optimizing Medal S for challenging datasets, particularly those involving instance segmentation with small lesions, significant foreground-background imbalances, or blurred boundaries, such as tumor data. Additionally, addressing datasets with a high number of classes and intricate anatomical relationships will be a priority. To align with clinical scenarios, we aim to develop robust solutions for diverse, complex datasets with numerous lesions, ensuring the model's applicability in real-world medical imaging tasks.

## 5    Conclusion

This study introduces Medal S, a novel medical image segmentation foundation model that achieves superior performance across diverse modalities. Its end-to-end trainable framework uniquely harmonizes spatial precision with semantic textual guidance by supporting native-resolution spatial and textual prompts. A key innovation is the channel-wise alignment of spatial prompts and text embeddings, which successfully mitigates inaccuracies from resolution mismatches—a critical limitation of text-only methods. Crucially, this parallel processing of spatial prompts yields a $10\times$ speedup for 24-class segmentation over sequential methods, with even greater efficiency gains for more classes. This advancement is further enabled by a lightweight 3D convolutional module for precise voxel-space refinement.

Medal S demonstrates superior efficiency and accuracy in multi-class 3D medical segmentation. On the five-modality average on the validation set, it outperforms SAT with a DSC of 75.44 (vs. 69.83), NSD of 77.34 (vs. 71.06), F1 of 38.24 (vs. 24.88), and DSC TP of 65.46 (vs. 46.97). Supporting up to 243 classes across CT, MRI, PET, ultrasound, and microscopy, Medal S offers two versatile prompting modes: text-only for self-refinement and hybrid for incorporating manual annotations. Future work will focus on improving robustness for complex, imbalanced datasets and small lesions, and refining the dynamic resampling strategy to address the current performance lag in ultrasound. Overall, Medal S represents a significant advancement for high-speed, memory-efficient, and accurate medical image segmentation.

**Acknowledgements** We thank all the data owners for making the medical images publicly available and CodaLab [24] for hosting the challenge platform.

**Disclosure of Interests.** The authors have no competing interests to declare that are relevant to the content of this article.

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

**Table 7.** Checklist Table. Please fill out this checklist table in the answer column. (**Delete this Table in the camera-ready submission**)

| Requirements | Answer |
| --- | --- |
| A meaningful title | Yes |
| The number of authors ($\leq 6$) | 6 |
| Author affiliations and ORCID | Yes |
| Corresponding author email is presented | Yes |
| Validation scores are presented in the abstract | Yes |
| Introduction includes at least three parts: background, related work, and motivation | Yes |
| A pipeline/network figure is provided | Figure 2 |
| Pre-processing | Page 8, 9, 10, 11 and 13 |
| Strategies to data augmentation | Page 14 |
| Strategies to improve model inference | Page 8, 9, 10 and 11 |
| Post-processing | Page 12 |
| Environment setting table is provided | Table 3 |
| Training protocol table is provided | Table 1 and 2 |
| Ablation study | Table 4, Table 5 and Figure 4 |
| Efficiency evaluation results are provided | Table 4 and Figure 4 |
| Visualized segmentation example is provided | Figure 3 and Figure 5 |
| Limitation and future work are presented | Yes |
| Reference format is consistent. | Yes |
| Main text $>=$ 8 pages (not include references and appendix) | Yes |