# OpenReview forum: "Medal S: Spatio-Textual Prompt Model for Medical Segmentation"
_thecvf.com/CVPR/2025/Workshop/MedSegFM — CVPR 2025 Workshop MedSegFM Submission_

### Official Review · Reviewer_JhtG · 2025-09-29
**Medal S: Spatio-Textual Prompt Model for Medical Segmentation**

**Rating:** 7
**Confidence:** 4

**Review:**

This paper introduces Medal-S, a novel medical image segmentation model that supports both textual and spatial prompts. The authors enhance the baseline architecture by implementing channel-wise alignment between volumetric prompts and text embeddings, effectively addressing spatial-text misalignment and enabling precise, simultaneous refinement. Additionally, they propose a dynamic resampling strategy to mitigate target-patch ratio imbalance, which significantly improves the detection of small segmentation targets. Extensive experiments demonstrate the strong performance of the proposed method. The authors also release detailed code and methodology, contributing valuable resources to the research community. While the dynamic resampling module still requires further refinement to generalize across diverse modalities, the overall work presents a meaningful advancement and merits acceptance.

Strengths
-  The paper introduces a prompt encoder capable of extracting information from both spatial and textual prompts. Additionally, it proposes a spatial prompt generation mechanism that supports interactive applications and autonomous refinement in segmentation tasks.
- A core innovation is the dynamic resampling technique, which addresses the challenge of varying segmentation target sizes relative to fixed patch dimensions. The method adjusts voxel spacing based on the physical size of the smallest foreground connected component or the smallest class-specific target, improving segmentation accuracy for small structures.
- The authors provide thorough explanations of each module and release the implementation code, enhancing reproducibility and contributing valuable resources to the research community.
- The proposed model achieves superior results in both semantic and instance segmentation tasks, outperforming the baseline (Medal-S: 75.55 Dice vs. CAT: 68.68 Dice on semantic segmentation), demonstrating its effectiveness.

Weaknesses
- The model struggles with instance segmentation in modalities such as Microscopy and PET, particularly when dealing with overlapping structures. Further experiments and analysis are needed to explore this limitation. Including visualizations of failure cases would also help clarify the model’s shortcomings.
- While the text explains the methodology, Figure 2 remains confusing. The initial processing steps are unclear, and the flow of operations is difficult to follow. The authors are encouraged to: number each step in the figure and include brief titles or descriptions for each stage in Section 2 to improve readability, or provide an additional simplified overview figure of the pipeline.
- Although the final metrics show improvement, the individual contributions of modules such as spatial prompt generation, dynamic resampling, and spatial prompt refinement are not clearly demonstrated. Additional ablation studies would help quantify the impact of each component.
- The paper claims that the two-stage inference improves computational efficiency and segmentation accuracy, but lacks experimental evidence or theoretical justification. In practice, two-stage models often introduce greater computational complexity. Including a model complexity comparison table and runtime analysis would strengthen this claim.

---

> ### Author Rebuttal · Authors · 2025-11-04
>
> We thank Reviewer JhtG for the insightful and constructive feedback. Below, we address each concern point-by-point.
>
> 1.Regarding the weaker instance segmentation performance on Microscopy and PET modalities, we have added BiomedParse-V results on the validation set in Table 4, showing that Medal-S achieves performance comparable to the top model BiomedParse-V. On the test set in Table 6, Medal-S lags behind BiomedParse-V, likely due to distribution shifts between validation and test sets or inference timeouts on some test cases. We will investigate this further in future work.
>
> 2.Given the heterogeneous spacing, volume sizes, inference-time, and GPU-memory constraints across challenge datasets, we adopted modality-specific strategies, resulting in the multi-stage inference pipeline. We will continue to streamline and unify the pipeline in future iterations. Figure 2 illustrates the core components of our model, including the input and processing flow of textual and parallel spatial prompts.
>
> 3.To demonstrate the effectiveness of parallel spatial prompting, we added Figure 4, Figure 5, and Table 5. Table 4 now includes ablations with and without the first stage as well as with and without post-processing.
>
> 4.On latency/memory scaling with respect to the number of classes: parallel spatial prompting achieves minimal time complexity relative to class count, delivering over 10x speedup for 24-class segmentation compared to sequential approaches, whose runtime grows substantially. Memory footprint increases only marginally, making it suitable for practical deployment.
>
> 5.On generating refined outputs given accurate spatial prompts: when provided with more accurate spatial prompts (e.g., ground truth masks), the model produces significantly refined outputs, effectively suppressing small-scale noise, resolving category confusion, and enhancing spatial continuity.
>
> 6.On the cost/benefit of two-stage refinement: Table 4 now includes ablations with and without the first stage. Using coarse Stage-1 predictions as spatial prompts for Stage-2 yields consistent gains (avg. +1.69 DSC, +2.86 NSD), with up to +13.37 F1 on MRI—proving coarse-to-fine guidance crucial for precise multi-organ localization. Tables 4 and 4.1 further compare the use of stage-1 spatial prompts and post-processing. Figure 4 and Table 5 detail the impact of stage 1 on memory and inference time, confirming that parallel spatial prompting adds negligible overhead to runtime.

---

### Official Review · Reviewer_i2AK · 2025-09-29
**Medal-S Review: Elegant 3D Spatio-Textual Alignment, but Needs Broader Baselines, Data Transparency, and Compliance Clarification”**

**Rating:** 6
**Confidence:** 4

**Review:**

Summary

The paper proposes Medal-S, a promptable 3D medical segmentation system that fuses native-resolution spatial prompts with text prompts using a lightweight channel-wise alignment and a compact 3D refinement head. The pipeline adds dynamic resampling, a two-stage coarse→fine inference scheme, and probability-aware post-processing. Results versus CAT are promising. However, baseline coverage is too narrow for broad SOTA claims; core reproducibility details are missing; runtime/memory scaling for parallel multi-class inference is unreported; qualitative evidence is only 2D despite a 3D focus; ultrasound underperforms without analysis; and the stated use of H100 GPUs with a Shanghai affiliation requires an export-control compliance statement.

Evaluation
Quality

Strengths include a simple and effective channel-wise alignment that preserves native 3D spatial detail, true parallel multi-class prompting at voxel resolution, a thoughtful spatial-prompt generator that simulates imperfect cues, and practical engineering choices (dynamic resampling, two-stage refinement, probability-aware component filtering). Weaknesses are empirical: baselines are limited to CAT (with SAT discussed) and omit BiomedParse and strong interactive 3D families such as nnInteractive, SegVol, SAM-Med3D, VISTA3D, and CT-SAM3D; reproducibility is hindered by the absence of dataset counts, split protocol, seeds, epochs, sampling schedule, and checkpoint selection criteria; efficiency is not quantified as the number of classes increases or with/without two-stage refinement; qualitative figures are only 2D slices; ultrasound underperformance is not analyzed; and compute compliance is not documented.

Clarity

The high-level design, the channel-wise alignment idea, and the prompt-synthesis routine are conveyed clearly. Clarity would improve with a consolidated hyper-parameter table (thresholds, spacing bounds, crop inflation, component filtering), a precise definition of instance-level metrics, a compact class inventory with canonical and variant names used in the text mapping, true 3D qualitative figures (meshes or volume renderings) with surface-distance maps, and a small-compute recipe so readers can reproduce results on 1–2 GPUs.

Originality

Medal-S is differentiated by native-resolution, parallel spatio-textual prompting with a clean channel-wise alignment and a training-time prompt noise model tailored to multi-class 3D masks. It is close in spirit to CAT but simpler than heavy cross-attention variants, and distinct from text-only (e.g., SAT, BiomedParse) or purely interactive families (SegVol, SAM-Med3D).

Significance

A promptable, multi-class, native-3D approach is practically valuable for clinical workflows involving organs and lesions across modalities. The potential impact is high if the method’s efficiency and robustness are demonstrated more completely and if baseline comparisons broaden beyond CAT.

Pros and Cons

Pros

Clean, compute-light channel-wise alignment that preserves native 3D detail

Parallel, native-resolution multi-class prompting rather than sequential decoding

Robust spatial-prompt synthesis mimicking imperfect user or auto prompts

Practical system engineering: dynamic resampling, coarse→fine refinement, probability-aware component filtering

Cons

Baseline coverage too narrow for broad SOTA claims; missing BiomedParse and interactive 3D families

Reproducibility gaps: no counts, splits, seeds, epochs, sampling schedule, or checkpoint selection rule

No latency/memory scaling results versus number of classes; no cost/benefit of two-stage refinement

No true 3D qualitative visualizations; only 2D slices

Ultrasound underperformance lacks targeted analysis or mitigation

No export-control compliance statement despite the stated H100 setup and affiliation

Actionable Requests

Reproducibility appendix:
Provide per-modality and major class-subset volume counts for train/validation/test; the exact split protocol (fixed versus k-fold), patient-level separation and seeds; the number of epochs or a steps-to-epochs conversion; sampling rates across datasets; preprocessing (intensity windows, resampling bounds); and post-processing thresholds with a brief sensitivity table.

Baselines and scope:
Add a subset comparison against a modern text-prompt foundation model such as BiomedParse on a public CT/MR split and include at least one interactive 3D family baseline (e.g., SegVol or SAM-Med3D). If this is infeasible, narrow the claim precisely and explain why.

Efficiency and scaling:
Report wall-clock latency and peak memory on a reference GPU as the number of simultaneous classes increases (for example 1, 4, 8, 16, 32). Include with/without two-stage refinement and the effect of different resampling bounds.

3D qualitative evidence:
Include meshes or isosurfaces and volume renderings for representative CT, MR, PET, ultrasound, and microscopy cases. Add surface-distance heatmaps and show at least one overlapping hierarchy (organ and contained lesion) with a discussion of leakage control.

Ultrasound analysis:
Provide per-class Dice/NSD histograms, target-patch ratio statistics, spacing/crop ablations tailored to large homogeneous organs, and consider a simple hierarchical override rule where lesions supersede parent organ predictions when confident.

Ablation grid:
Quantify the marginal contribution of dynamic resampling, two-stage refinement, iterative feedback, spatial-prompt synthesis, and post-processing. Compare the proposed alignment to simpler concatenation and to cross-attention or low-rank learned mixing; consider a class-competition mechanism to reduce bleed-through.

Text robustness:
Evaluate zero-shot synonyms not in the curated mapping, ambiguous prompts, and multilingual/noisy phrasing; report degradation and demonstrate the benefit of the mapping layer.

Export-control compliance:
Add a short statement describing compute location (and cloud region if applicable), ownership and jurisdiction of the hardware, license status where relevant, or confirmation that compute was performed outside restricted jurisdictions or with non-restricted accelerators. Correct any tables if different hardware was actually used and include minimal logs/configs.

Decision Guidance

If the authors add data/splits/epochs disclosure, 3D visual evidence, efficiency scaling, and at least one cross-family baseline comparison (or precisely narrow claims), I will support acceptance with shepherding. If these remain missing, a major-revision verdict is appropriate. The compliance statement is a required transparency item but should not be the sole basis for rejection; the area chair or editor can evaluate it against venue policy.

Scores (provisional)

Quality: 6.5/10 (rising to ~7.5 with broader baselines, efficiency plots, and 3D visuals)
Clarity: 7/10
Originality: 7.5/10
Significance: 7/10

---

> ### Author Rebuttal · Authors · 2025-11-04
>
> We thank Reviewer i2AK for the insightful and constructive feedback. Below, we address each concern point-by-point.
>
> 1. On the claims of state-of-the-art performance and experimental analysis:
> Claims & Analysis: We have replaced "state-of-the-art" claims with "superior/excellent" and added validation/test results. Due to fundamental differences in our training paradigm, we avoid direct comparison with interactive methods for now. New figures and a table demonstrate the effectiveness of our parallel spatial prompting.
>
> Latency/Memory Scaling: Our parallel prompting method achieves minimal time complexity relative to the number of classes, offering a >10x speedup for 24-class segmentation over sequential approaches, with only a marginal memory increase.
>
> Refined Outputs: When provided with accurate spatial prompts (e.g., ground truth masks), the model produces significantly refined outputs by suppressing noise, resolving category confusion, and improving spatial continuity.
>
> Two-Stage Refinement: Ablation studies in Table 4 confirm the benefits of our two-stage inference. Using coarse predictions from stage one to guide stage two yields substantial performance gains, proving this coarse-to-fine guidance is essential for precise localization.
>
> 2. On the reproducibility gaps regarding counts, splits, seeds, epochs, sampling schedule, and checkpoint selection rule:
> Relevant settings are detailed in Section 3.2, Implementation Details. Our training code is largely built upon SAT and nnU-Net. We have open-sourced all training hyperparameters and full training and inference code at: https://github.com/yinghemedical/Medal-S
>
> 3. On the lack of true 3D qualitative visualization and only showing 2D slices:
> We have added 3D visualizations in Figure 5, which provides a qualitative comparison of Medal-S with different spatial prompt configurations.
>
> 4. On the underperformance in ultrasound and the lack of targeted analysis or mitigation:
> Regarding ultrasound underperformance: A target-patch imbalance was identified. During training, high-resolution volumes were used, but at inference, large target structures often occupied most of the image, while sampled patches covered only a small portion. This issue can be mitigated by lowering resolution. We plan to develop a more robust dynamic resampling strategy in future work.
>
> 5. On the computational resources:
> The computational machines with H100 GPUs provided by the co-authors are not located in Shanghai.

---

### Official Review · Reviewer_kHNL · 2025-10-12
**Medal S fuses native resolution spatial prompts with text prompts via channel-wise alignment, predicts all classes in parallel, and self-refines using its own outputs. It adds dynamic resampling, coarse to fine two-stage inference, and probability-aware post-processing.**

**Rating:** 7
**Confidence:** 4

**Review:**

Medal S is an iterative 3D segmentation framework that fuses native-resolution spatial prompts with text prompts through a channel-wise alignment mechanism. It performs parallel prediction for all classes simultaneously, then self-refines by feeding its own output back as an improved spatial prompt for subsequent iterations. It reports a strong gain over baseline methods specially in CT and MR modalities on the validation set. However, test set results are not reported on the paper.

**Strengths :**

The model performs a channel-wise alignment between per-class text embeddings and full-resolution volumetric prompts. By incorporating lightweight 3D convolutions and avoiding the use of downsampled mask tokens, this approach preserves fine voxel detail and reduces spatial misalignment.

Medal S processes all classes in parallel, eliminating the need for sequential, per-class loops. This parallel design not only boosts efficiency but also allows the model to leverage inter-organ context.


**Weakness :**

Ultrasound underperformance is under-explained. The paper attributes US drops to “target–patch ratio” issues, but this isn’t convincingly isolated from other ultrasound-specific factors. If target–patch imbalance were the dominant driver, we’d expect similar failures in other modalities/classes and could be supported by ablation study.

The system looks heavy and hard to reproduce. Training with multi-stage pipelines and iterative refinement, and using H100s GPUS raises barriers for typical academic labs and clinics. Given the compute footprint, the paper could offer a smaller, easier-to-deploy path.

Post-processing may not generalize across datasets as it uses fixed thresholds for connected component.

---

> ### Author Rebuttal · Authors · 2025-11-04
>
> We thank Reviewer kHNL for the detailed and constructive feedback. Below we address each concern point-by-point.
>
> 1. Test set results
>    We have added the final test set performance in Section 4.4 (Results on final testing set) and Table 6.
>
> 2. Ultrasound underperformance and target–patch ratio
>    We agree that the explanation in the original submission was insufficient. During training we used native-resolution ultrasound volumes. Only at inference did we observe that target structures often occupy a very large proportion of the entire image, while the sampled patch covers only a small portion, leading to severe target-patch imbalance. Lowering the resolution during training/inference can mitigate this issue. We plan to develop a more robust dynamic resampling strategy in future work.
>
> 3. System complexity and reproducibility
>    Given the heterogeneous spacing, volume sizes, inference-time, and GPU-memory constraints across challenge datasets, we adopted modality-specific strategies, which resulted in the multi-stage inference pipeline. We will continue to streamline and unify the pipeline in future iterations.
>    During the competition we used large batch and patch sizes on multi-H100 GPUs to accelerate training and maximize performance. However, the model can be trained on 40 GB or even 24 GB GPUs by reducing batch and patch sizes. Inference memory is low: even with 24 classes and parallel 3D spatial prompts, only ~12 GB is required (see Figure 4).
>
> 4. Post-processing generalization
>    Additional ablation (Table 4) shows that the current connected-component post-processing provides limited gains. We will explore more adaptive and dataset-agnostic post-processing in future work.

---

### Decision · Program_Chairs · 2025-11-12

Accept